# Lipids: Key Players That Modulate α-Synuclein Toxicity and Neurodegeneration in Parkinson’s Disease

**DOI:** 10.3390/ijms21093301

**Published:** 2020-05-07

**Authors:** Akio Mori, Yuzuru Imai, Nobutaka Hattori

**Affiliations:** 1Department of Neurology, Juntendo University Graduate School of Medicine, Tokyo 113-8421, Japan; a-mori@juntendo.ac.jp; 2Department of Research for Parkinson’s Disease, Juntendo University Graduate School of Medicine, Tokyo 113-8421, Japan

**Keywords:** phospholipids, PLA2G6, Parkinson’s disease, Lewy bodies, NBIA

## Abstract

Parkinson’s disease (PD) is the second most common neurodegenerative disease; it is characterized by the loss of dopaminergic neurons in the midbrain and the accumulation of neuronal inclusions, mainly consisting of α-synuclein (α-syn) fibrils in the affected regions. The prion-like property of the pathological forms of α-syn transmitted via neuronal circuits has been considered inherent in the nature of PD. Thus, one of the potential targets in terms of PD prevention is the suppression of α-syn conversion from the functional form to pathological forms. Recent studies suggested that α-syn interacts with synaptic vesicle membranes and modulate the synaptic functions. A series of studies suggest that transient interaction of α-syn as multimers with synaptic vesicle membranes composed of phospholipids and other lipids is required for its physiological function, while an α-syn-lipid interaction imbalance is believed to cause α-syn aggregation and the resultant pathological α-syn conversion. Altered lipid metabolisms have also been implicated in the modulation of PD pathogenesis. This review focuses on the current literature reporting the role of lipids, especially phospholipids, and lipid metabolism in α-syn dynamics and aggregation processes.

## 1. Introduction

Several neurodegenerative diseases, including Parkinson’s disease (PD), PD dementia (PDD), dementia with Lewy bodies (DLB), and multiple system atrophy (MSA), are described as synucleinopathies, which are characterized by the progressive accumulation of fibrillized α-synuclein (α-syn) in the affected regions [1,2]. Lewy bodies (LBs), which consist mainly of fibrillized α-syn, is the main pathological hallmark of PD, PDD, and DLB [2]. PD is the most common synucleinopathy in the elderly [1]. It is symptomatically characterized by motor disturbances, such as body bradykinesia, resting tremor, rigidity, and postural instability, as well as by non-motor features, such as failures in the autonomic nervous system, and dementia [1]. The early prominent death of dopaminergic neurons in the substantia nigra causes motor disturbance and core pathogenesis in PD [3]. These symptoms are challenged by symptomatic treatments, mainly with the dopamine precursor levodopa and dopamine receptor agonists.

Alpha-synuclein is expressed predominantly in neurons. However, α-syn forms aggregations in glial cells in these synucleinopathies, including PD [4,5]. MSA is a sporadic, adult-onset neurodegeneration disease classified in atypical parkinsonian syndromes with α-syn-positive inclusions observed predominantly in the cytoplasm of oligodendrocytes [6]. These inclusions are labeled (oligodendro)glial cytoplasmic inclusions (GCIs) [7], and distinct α-syn strains or pathological forms of α-syn have been hypothesized to accumulate in GCIs [8]. Although the physiological function of α-syn in glial cells is not fully understood, α-syn is suggested to activate the innate immune system, in which microglia and astrocytes are involved, and to regulate astrocytic fatty acid metabolism [9,10].

Genetic studies provide a compelling evidence that quantitative and qualitative variation of α-syn protein modulates the risk of PD and MSA development. Point mutations in the *SNCA* gene encoding α-syn are associated with autosomal dominant PD [11,12,13,14,15]. Multiplications of the *SNCA* gene are also linked to familial PD [16] and *SNCA* polymorphisms are associated with an increased risk of sporadic PD and MSA [17,18,19].

Recent high-resolution histochemical techniques have revealed that LBs contain α-syn fibrils and various cellular materials, including the mitochondria, lysosomes, and membrane lipids [20]. The brain is one of the richest in lipid content among the body tissues [21,22]. A variety of lipids exist in the brain, including fatty acids, triacylglycerols, phospholipids, sterols, and glycolipids. These lipid species are utilized for energy metabolism, protein modification, signal mediators, and biomembrane and organelle functions. In both the physiological and pathological aspects of α-syn, the roles of lipids in biomembranes are an important factor to consider. The binding of α-syn to vesicle membranes appears to be necessary for vesicle function through the assembly of the soluble N-ethylmaleimide-sensitive factor-attachment protein receptor (SNARE) complex in the presynaptic terminals. The altered affinity of α-syn toward lipid membranes could affect aggregation process. The findings of *PLA2G6*, *VPS13C*, *LIMP2*, *GBA1,* and *GALC* as the genes responsible for or genetic risks of PD with prominent LB deposition strengthen the concept that lipids are indeed involved in the aggregation and propagation of α-syn [23,24,25,26,27]. These genes are suggested to regulate metabolism, transport, and degradation of lipids [24,25,27,28,29].

Unfortunately, there is no definitive therapy for PD. Understanding the pathogenic mechanisms associated with α-syn aggregation is crucial to develop effective disease-modifying therapies [30]. This review focuses on the literature linking α-syn function (or misfunction) mainly with phospholipids, which is a major constituent molecule of biomembranes and membrane structures, as well as how these interactions participate in the physiological function and the pathogenesis in PD. Finally, we would like to discuss potential therapeutic targets for lipid modification in PD.

## 2. Properties of the α-Synuclein Protein

The synuclein family consists of α-, β-, and γ-synuclein, but no disease-associated mutations have been reported in β- and γ-synuclein [31]. Alpha-synuclein is a small protein with 140 amino acids that were originally identified together with other homologues from the cholinergic synaptic vesicles of the electric fish *Torpedo californica* in 1988 [32]. Although α-syn is ubiquitously distributed in red blood cells, lymphocytes, muscle, kidney, heart, and lung [33,34,35,36,37], it is also very abundant in the brain tissue, accounting for about 1% of the total protein in the rat brain [38].

In the brain tissue, α-syn is expressed in neurons and highly localized to the presynaptic terminals [39]. Indeed, synuclein was named because of its localization at the synapse and nuclear envelope. Figure 1 depicts the protein structure of α-syn. Alpha-synuclein is divided into three regions, including (1) the-N-terminal repeat region (residues 1–60); (2) the highly hydrophobic central portion (residues 61–95); and (3) the acidic C-terminal region (residues 96–140). The N-terminal region contains 11-amino acid imperfect repeats with a highly conserved hexamer motif (XKTKEGVXXXX) that form amphipathic α-helical features. The central portion is known as the non-amyloid-component (NAC) domain, which was isolated from amyloid plaques of the brains of patients with Alzheimer’s disease (AD). AD is the most common neurodegenerative disease associated with the accumulation of β-amyloid and some AD cases also exhibit an LB pathology. The NAC domain is necessary for α-syn fibrillation. Indeed, the deletion of residues 71–82 within the NAC domain caused α-syn to lose its aggregation property in a transgenic *Drosophila* model [40]. The N-terminal region of α-syn binds to artificial lipid membranes that mimic the composition of the synaptic vesicles [41,42,43,44,45].

However, like other neurodegenerative diseases with specific protein inclusions, the processes of α-syn aggregation and LB formation and their roles in neurodegeneration remain unclear. Some studies have shown that a monomer form of α-syn changes into pathogenic β-sheet-rich small fibrils (seeds), subsequently elongating into twisted amyloid fibrils, and finally forming LBs [3]; this process appears to disrupt the vesicular transport and organelle functions of the mitochondria, endoplasmic reticulum (ER), and lysosomes, eventually leading to neuronal death [3].

## 3. N-Terminal Amphipathic Helices of α-Synuclein Bind with Phospholipids

Alpha-synuclein has a high degree of sequence homology with apolipoproteins, which function in lipid transport as structural components of lipoprotein particles, enzymatic cofactors, and ligands for cell-surface receptors [46]. Amphipathic helices, first found in the amino acid sequences of plasma apolipoprotein, are a secondary structural motif involved in lipid membrane association. A helical wheel analysis identified α-syn as having five potential amphipathic α-helices in the N-terminal region (Figure 2); four of these five theoretical helices shared the class A2 lipid-binding motif properties, which have been best defined and correlated with the highest lipid affinity and structurally similar to the class A2 lipid-binding motif in apolipoproteins [39,47]. The characteristic N-terminal 11-residue sequence (consensus XKTKEGVXXXX) is repeated seven times in α- and γ-synuclein and six times in β-synuclein.

The N-terminal binding residues of α-syn to phospholipids are highly conserved among species, in contrast to the less conserved C-terminal region [32]. The conserved N-terminal region suggests that α-syn, like apolipoproteins, is capable of reversibly binding to the lipid membranes. Alpha-synuclein exists in an equilibrium between a soluble and a membrane-bound state; the membrane-bound form accounts for approximately 15% of the total α-syn in the rat brain [48] (Figure 2). In fact, α-syn directly binds to the synaptic vesicles at the presynaptic terminals in the zebra finch [39], rat [32,49,50], and mouse [51], and is colocalized with β-syn in the human brain [37,52].

## 4. Association of α-Synuclein with Lipid Membrane

Binding assays of α-syn to lipid membranes have been performed by artificial liposomes. Davidson et al. first showed that α-syn binds to liposomes containing acidic phospholipids [47]. Many previous studies have shown that negatively charged phospholipids, including phosphatidylserine (PS) and phosphatidic acid (PA), as well as other lipids, such as sphingolipids and fatty acids, affect the binding affinity of α-syn to membranes and/or the α-syn aggregation by the membranes [28,41,54,55,56,57,58].

Phospholipids are the main components of biomembranes and play essential roles in cellular functions, including apoptosis, protection against oxidative damage, generation of secondary messengers, and regulation of enzyme activities [22]. Phospholipid diversity is attributed to the combination of head groups with hydrocarbon chains that vary in fatty acid length, double bond number, and position. Biochemical studies have shown that vesicles containing acidic phospholipids, such as PA and PS, but not neutral phospholipid phosphatidylcholine (PC) and phosphatidylethanolamine (PE) alone, interact with α-syn [41,47,57,59]. These results might be explained by the effects of the electrostatic head group interaction as well as by increased packing defects in the membrane because the head groups of PA and PS are cone-shaped lipids (small head groups relative to the large acyl chains).

The liposomes of the phospholipid compositions that show an increasing affinity for α-syn are similar to those of synaptic vesicles. Synaptic vesicles contain approximately 1%–3% PA of the total phospholipids, whereas PC and PS are more abundant, reaching 36% and 12%, respectively [21,60]. Moreover, the lipid composition of PA and PS in synaptic vesicles are more abundant than those in the mitochondria and lysosomes [61,62]. Aspects other than head group composition are also important when considering phospholipid properties. Poly-unsaturation of acyl chains is required for the binding of α-syn with membranes [57], whereas α-syn aggregation is accelerated by the composition of lipids with shorter saturated acyl chains [28,56]. Acyl chain length affects the membrane fluidity, which has a crucial role in preventing α-syn aggregation [28,63].

Vesicle size affects the membrane curvature and affinity of α-syn. Alpha-synuclein tends to bind to small unilamellar vesicles (SUVs) rather than to multilamellar vesicles in the same lipid component [47,64]. Moreover, it has been shown that post-translational modifications of α-syn are another factor to modulate its lipid binding. The acetylated forms of α-syn change its affinity to the membrane, especially to PC micelles and SUVs with a high curvature [63]. A nuclear magnetic resonance analysis in the mammalian cells suggested that the disordered nature of monomeric α-syn was preserved in non-neuronal and neuronal cells and that the α-syn N-terminus was acetylated under physiological conditions [65].

## 5. Functions of α-Synuclein at Synapses

Although the physiological functions of α-syn are not fully understood, the abundance of α-syn at the presynaptic terminals suggests that α-syn plays a role in synaptic functions. In the presynaptic terminals, α-syn has been shown to colocalize with presynaptic proteins, such as synapsin I, synapsin III, synaptotagmin, and synaptophysin [66].

Synaptic vesicle exocytosis for neurotransmitter release is regulated by action potentials in the presynaptic terminals. Action potential generation leads to a Ca^2+^ influx into the presynaptic terminal via Ca^2+^ channels. This Ca^2+^ influx promotes the assembly of the Ca^2+^ sensor synaptotagmin-1 and the SNARE [67]. SNARE proteins are divided into the vesicle-associated (v-) SNARE and the target plasma membrane (t-) SNARE proteins; both proteins form a helical coiled-coil complex to execute a synaptic vesicle fusion [68]. Synaptic vesicles in the closest vesicle pool to the active zone, the readily releasable pool, have been hypothesized to be docked and fused to the presynaptic plasma membrane at the active zone. After the readily releasable pool is depleted, continued release occurs from the secondary releasable vesicle pool, the recycling pool [69].

Mice overexpressing α-syn exhibit a synaptic function disturbance in contrast to the subtle effects seen in α-syn knockout mice [70,71,72,73]. For instance, α-syn overexpression reduces vesicles in the readily releasable pool and the recycling pool due to an inhibition in the re-clustering of synaptic vesicles after endocytosis [71]. There are six α-syn point mutations known to be associated with familial PD (A30P, E46K, H50Q, G51D, A53E, and A53T), which are in the N-terminal membrane-binding domain [11,12,13,14,15]. The overexpression of A30P α-syn, which has been suggested to decrease its affinity toward membranes [51], shows no defect in a synaptic vesicle exocytosis [71]. In contrast to A30P, overexpression of A53T and E46K, which retain the membrane-binding affinity [54,74], disturb the transmitter release [71]. These results indicate that the N-terminal membrane-binding domain property of α-syn affects synaptic vesicle dynamics. Indeed, native α-syn promotes the fusion and clustering of vesicles upon binding to the synaptic plasma membrane through the v-SNARE VAMP2 (synaptobrevin-2) (Figure 3) [75,76].

The C-terminal region of α-syn directly binds to the N-terminal of VAMP2 on the synaptic vesicle surface in vivo [75,76]. Both overexpressed and endogenous α-syn accelerate exocytosis and fusion pore dilation upon a synaptic vesicle docking at the plasma membranes in the active zones [78] (Figure 3). The binding of α-syn with VAMP2 finally facilitates the assembly of chaperones, such as cysteine string protein-α (CSPα), heat-shock cognate 70 (Hsc70), and small glutamine-rich tetratricopeptide repeat-containing protein (SGT). CSPα contains a DNA-J domain, which is characteristic of the heat shock protein 40-like co-chaperones [79,80]. CSPα activates the Hsc70 ATPase activity and forms a tertiary complex with Hsc70 and SGT on the synaptic vesicle. The tertiary complex binds to a complex with the plasma membrane SNARE protein syntaxin-1 and SNAP-25 on the target membrane in the active zones [77]. In contrast to the abovementioned hypothesis, an in vitro study demonstrated that dopamine-induced α-syn oligomers blocked the SNARE-mediated vesicle docking by binding to the N-terminal region of VAMP2 [81]. The opposite results of α-syn oligomers in the vesicle docking could come from the physiological and pathological forms of oligomers generated in different experimental settings.

The loss of CSPα in *Drosophila* and mice results in the impairment of synaptic functions and neurodegeneration [82,83]. Mutations in the CSPα-encoding *DNAJC5* genes cause an adult-onset neuronal ceroid lipofuscinosis (ANCL), also known as the autosomal dominant Kufs disease and Parry disease. ANCL is a very rare hereditary neurodegenerative disorder that presents with generalized epilepsy, movement disorders, and progressive dementia [84]. Although CSPα and α-syn appear to act differently at synapses, an α-syn overexpression rescues the neurodegeneration observed in CSPα knockout mice, providing a clue to the physiological roles of α-syn [75,85].

## 6. Roles of Membrane In α-Synuclein Aggregation Pathogenesis

Natively-unfolded α-syn has been hypothesized to undergo a conformational change into oligomers and a β-sheet-rich amyloid conformation that is associated with aggregation, fibril formation, and, finally, an LB formation [3]. In this pathogenic pathway, lipid membranes are likely to regulate the conformational changes of α-syn. Noteworthy, all six α-syn point mutations are in the N-terminal membrane-binding domain [11,12,13,14,15]. Several studies have reported that α-syn mutant affinity to membranes decreases in the A30P, A53E, and G51D mutations, increases in the E46K mutation, but remains the same as that in the wild-type in the A53T mutation; the H50Q mutation increases α-syn aggregation with subtle effects on the lipid-binding property (Table 1) [41,51,54,59,64,86,87,88,89], whereas the A53T and E46K mutations tends to aggregate and form pronounced twisted fibrils in vitro in the absence of membranes [89]. Thus, there should be two parameters to consider α-syn aggregation in terms of the effects of PD-associated mutations: Predisposition to β-sheet conformation and affinity to lipids. In contrast to the α-syn N-terminal regions, the C-terminal 103-140 aa in α-syn minimally affect lipid binding, or do not show any conformational change in the presence of phospholipids [41].

The issue of whether an α-syn aggregation occurs in its lipid-bound form or an unbound form continues to be debated [28,48,90,91,92,93,94]. Circular dichroism spectroscopy analysis revealed that vesicles containing acidic phospholipids (e.g., PA, PS, or PI) induced a conformational change of α-syn from random coils to > 70% α-helices, whereas vesicles containing PC alone did not affect the secondary structure. This secondary structure shift is attenuated in the A30P mutant [41]. An electron microscopy study revealed the human α-syn overexpression in mice resulted in extensive tubulovesicular structures appearing at the presynaptic terminals [95]. Moreover, LBs in the brain tissues from PD patients reportedly contain membrane fragments, vesicles, and organelles, including the mitochondria and lysosomes [20]. These findings support the idea that an α-syn aggregation promotes the disruption of membrane homeostasis or trafficking in neurons. Mitochondria-associated endoplasmic reticulum membranes (MAMs) are characterized as sites for an autophagosome formation, mitochondrial fission, Ca^2+^ transfer from the ER to mitochondria, and transport of phospholipids, fatty acids, and cholesterol between the ER and mitochondria [96,97,98]. A cultured cell study showed that a wild-type α-syn overexpression increased the number of MAMs, leading to an increased mitochondrial Ca^2+^ uptake from the ER to sustain mitochondrial function [99]; another study showed that A30P and A53T α-syn reduced its association with MAMs, promoting mitochondrial fragmentation [100]. Mechanistically, A30P showed the lower localization in MAM because of its lower affinity toward the membrane, while A53T showed a reduction in both membrane binding and the total protein expression compared with the wild type [100].

A prion-like spreading of the α-syn pathology has been proposed owing to the finding that LBs were detected in the grafted fetal mesencephalic cells of PD patients [101,102,103]. Several animal and cell model studies support the cell-to-cell transmission of α-syn [104,105,106,107,108,109]. Moreover, the spreading of α-syn pathology is closely correlated with disease progression and the pathological staging of PD [1,110,111]. Evidence from postmortem studies suggests that the α-syn pathology may spread from the enteric nervous system and propagate to the brain through the vagus nerve [110]. Animal studies support this new concept by showing the gut-to-brain spreading of α-syn [112,113].

The N-terminal truncation of α-syn (lacking its membrane-binding domain), results in an increased seeding activity and propagates efficiently in the mouse brain compared to a full-length α-syn [114]. Another piece of evidence for the importance of membranes in α-syn propagation is that aging alters the membrane fluidity via an increase in cholesterol and phospholipids with shorter acyl chains [115,116,117]. An in vitro study indicated that cholesterol reduced the binding of N-acetyl α-syn to 1,2-dioleoyl-sn-glycero-3-phosphocholine (DOPC) [63]. These observations support the possibility that the disruption of α-syn binding to the membrane also influences α-syn propagation.

Many neurotoxic mechanisms mediated by α-syn in conjunction with lipids have been proposed. Alpha-synuclein aggregation is triggered by the disruption of its affinity to the membrane [28], or induced by the pore-like morphology of the α-syn on the membranes [118]; the membrane disruption results in increased intracellular reactive oxygen species and a reduction in the mitochondrial activity [119]. Increasing membrane curvature and expansion by α-syn impairs membrane integrity [120,121]. These models should be validated by further studies.

## 7. Phospholipases Modulate the α-Synuclein Pathology

Phospholipases are a group of enzymes that hydrolyze phospholipids and regulate membrane homeostasis. Phospholipase classification is based on their site of action (Figure 4). Phospholipase A (PLA) hydrolyzes ester bonds, classified according to the position of the *sn*-1 (for PLA1) or the *sn*-2 (for PLA2) ester bonds [122]. Phospholipase B (PLB) hydrolyzes both ester bonds. Cleavage of the glycerophosphate bond is catalyzed by phospholipase C (PLC), whereas the removal of the base group is catalyzed by phospholipase D (PLD). Two classical PLD isoforms, PLD1 and PLD2, which are 50% identical in mammalian protein sequences, catalyze hydrolysis of the PC headgroup to generate PA and free choline.

PLD1 and PLD2 are membrane-bound lipases linked to a neurotransmitter release [123]. PLD2 is localized in the plasma membrane, especially in the detergent-insoluble membrane microdomains that contain the caveolar coat marker proteins caveolin-1 and caveolin-2 [124,125]. In addition, PLD2 interacts with α-syn and its enzymatic activity is inhibited by α- and β-syn in vitro [126]. Postmortem studies of DLB patients showed a reduction in the PLD1 activity and expression levels while PLD1 prevented the accumulation and cytotoxicity of α-syn by activating an autophagic flux [127]. Although PA, to which α-syn preferentially binds, is an intermediate generated in phospholipid biosynthetic pathways, PA itself contributes to the intracellular membrane traffic and exocytosis [41,47,54,55,128]. Several studies suggest that α-syn could affect the PA levels by inhibiting the PLD activity to exert a widespread influence over different cellular functions, including vesicle trafficking, membrane curvature, lysosomal activity, and PA-dependent signaling [129].

PLA2 enzymes catalyze the hydrolysis of the *sn*-2 ester bond of phospholipids, generating free fatty acids and lysophospholipids. The PLA2 family includes three major subgroups: (1) the secretory PLA2 (sPLA2), Ca^2+^-requiring secretory enzymes; (2) the cytosolic PLA2 (cPLA2), enzymes for arachidonic acid; and (3) the Ca^2+^-independent PLA2 (iPLA2), which play a major role in phospholipid remodeling [130].

*PLA2G6* (*iPLA2-VIA/iPLA2β*) has been reported as the causative gene for an autosomal recessive form of PD (*PARK14*), infantile neuroaxonal dystrophy (INAD), and neurodegeneration with brain iron accumulation (NBIA) [131,132]. All three diseases are caused by mutations in the *PLA2G6* gene and show a widespread LB pathology [23,133]. This evidence gives us a clue to understand the molecular mechanism of α-syn aggregation by an altered phospholipid metabolism. *PLA2G6* knockout mice exhibit an increased α-syn expression in the neurons as well as phosphorylated α-syn accumulation in damaged mitochondria [134]. Increased lipid peroxidation and mitochondrial damage have been reported in *PLA2G6-*deficient *Drosophila* [135]. The loss of *PLA2G6* in *Drosophila* results in dysregulation of the ceramide metabolism, leading to the inhibition of the retromer (a protein complex involved in the retrograde transport of membrane proteins in the endosomal-Golgi recycling pathway) and lysosomal functions [136], as well as an age-dependent shortening in the acyl-chain length in brain phospholipids due to membrane remodeling pathway disruption [28]. The shortening of the phospholipid acyl chains alters the membrane fluidity, lipid packing, and curvature. These changes affect the synaptic vesicle properties and decrease the affinity of α-syn toward the synaptic vesicles, resulting in α-syn aggregation [28].

cPLA2 selectively hydrolyzes arachidonyl phospholipids in the *sn-2* ester bond, releasing arachidonic acid. Synaptic damage by α-syn reportedly activates cPLA2, leading to further synaptic damage [137].

## 8. Other Lipid-Related Genes That Affect the α-Synuclein Properties

LB pathology is also observed in lysosomal storage disorders, such as Gaucher disease. Gaucher disease is caused by mutations in the *GBA1* gene, which encodes a lysosomal enzyme, glucocerebrosidase (GCase); *GBA1* mutations are a major risk factor for PD [138]. GCase hydrolyzes glucosylceramide (GluCer) into glucose and ceramide in the lysosomes [139]. GCase deficiency leads to the accumulation of defective lysosomes and results in an impaired autophagic clearance of α-syn [29]. Indeed, α-syn levels are associated with GCase deficiency in animal models and in humans [140,141,142,143,144]. Although the mechanism of α-syn aggregation induced by a GCase deficiency is not fully understood, two major mechanisms have been proposed: the increased levels of α-syn due to the lysosomal dysfunction [145,146] and the acceleration of α-syn aggregation by the accumulated GluCer [29,147] or glucosylsphingosine, another substrate of GCase [148]. The finding of a late endosomal-lysosomal flippase, ATP10B, as a risk gene for PD and dementia with LBs, strengthened the possibility of GluCer involvement in α-syn turnover [149]. ATP10B translocates GluCer and PC towards the cytosolic membrane side in endosomal–lysosomal vesicles, and the loss of ATP10B activity results in lysosomal dysfunction [149].

Similar to *PLA2G6, C19orf12* is a causative gene of NBIA with an LB pathology [150,151]. C19orf12 was originally reported as a mitochondrial membrane protein and later reported to be localized in the ER and MAM, as well as the mitochondria [28,152]. C19orf12 has been proposed to contribute to the mitochondrial membrane remodeling though lipid metabolism regulation [131]. Overexpression of C19orf12 suppresses α-syn aggregation by *PLA2G6* loss in a *Drosophila* model, suggesting that C19orf12 compensates for the phospholipid remodeling regulated by *PLA2G6* [28].

## 9. Conclusions

A series of studies imply that disruption of the lipid metabolism is part of the common pathogenesis underlying the α-syn aggregation process and neurodegenerative diseases, including PD, NBIA, and other LB diseases. The α-syn seeds for propagation are suggested to come from the intestinal nerve plexus [112,153]. The transition of α-syn to seeding should be subjected to lipid constituents in diet. An altered lipid metabolism affects the membrane property and could stimulate α-syn propagation through vesicular transport, such as an exosome [154]. The determination of risk lipids in this context is a challenging problem for the future. In the substantia nigra of PD patients, male-specific reduction in PC, PE, and PI species and elevated activities of phospholipid biosynthetic enzymes were reported [155], whereas a reduction in GCase activity was observed [145]. These observations might be associated with dopaminergic neurodegeneration and LB pathology. Moreover, in addition to the anterior cingulate cortex and amygdala, a variety of lipid species change in the visual cortex where the LB pathology is rarely observed, indicating that changes in lipids precede an LB deposition [156]. The concept of control of α-syn conformation by lipid regulation is a promising candidate for disease-modifying therapy in PD [30]. Indeed, an antimicrobial agent, squalamine [91], and a small molecule, NPT200-11 [157], which have advanced into a clinical trial (NCT02606682), successfully prevent α-syn aggregation by modulating the α-syn-binding affinity to membranes. Squalamine from dogfish sharks was found to displace α-syn from the surface of lipid membranes by masking α-syn lipid-binding sites. This molecule is reported to inhibit aggregation of α-syn [91]. The dietary manipulation of fatty acids over a long time period could change brain lipid components to prevent α-syn aggregation [28]. Further research to obtain exact information on the lipid species responsible for preventing an α-syn aggregation and subsequent disease development is warranted.

## Figures and Tables

**Figure 1 ijms-21-03301-f001:**
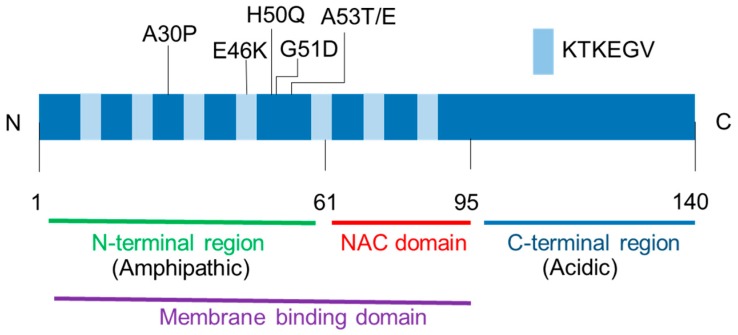
Alpha-synuclein protein structure. The N-terminal region and the non-amyloid-component (NAC) domain have seven imperfect KTKEGV repeats that contribute to membrane binding. All missense mutations linked to the familial Parkinson’s disease (A30P, E46K, H50Q, G51D, A53T, and A53E) are located in the N-terminal region.

**Figure 2 ijms-21-03301-f002:**
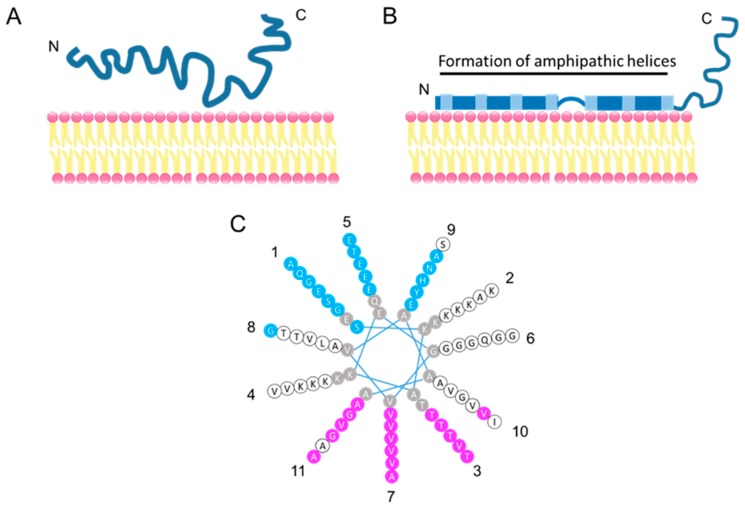
Model of membrane-bound α-syn. (**A**) Natively unfolded α-syn in solution. (**B**) Membrane-bound α-syn. The N-terminal region of α-syn folds into an amphipathic helical structure upon membrane contract. (**C**) The helical wheel model of the α-syn repeated region (9–89 aa). Residues in pink are hydrophobic and buried inside membranes when α-syn binds to membranes, residues in blue are exposed to the hydrophilic side (i.e., the cytoplasm), residues in white are on neither the membrane-exposed side nor on the solvent-exposed side, and residues in gray are not tested by continuous-wave electron paramagnetic resonance [53]. Eleven repeated lysine residues contribute to hydrogen bonding with acidic phospholipids. Numbers around the helical wheel indicate a repeat position. Adapted from [53].

**Figure 3 ijms-21-03301-f003:**
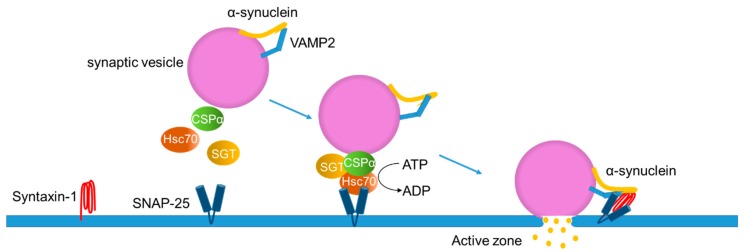
Proposed model for the physiological functions of α-synuclein in exocytosis. Cysteine string protein-α (CSPα) forms a complex with the heat-shock cognate 70 (Hsc70) and small glutamine-rich tetratricopeptide repeat-containing protein (SGT) on the synaptic vesicle. The tertiary complex binds to SNAP-25, a protein localized on the target membrane, to promote formation of the SNARE complex. The C-terminal region of α-syn binds to VAMP2 on the synaptic vesicle, assisting in the SNARE complex assembly together with the plasma membrane SNARE protein, syntaxin-1. Adapted from [77].

**Figure 4 ijms-21-03301-f004:**
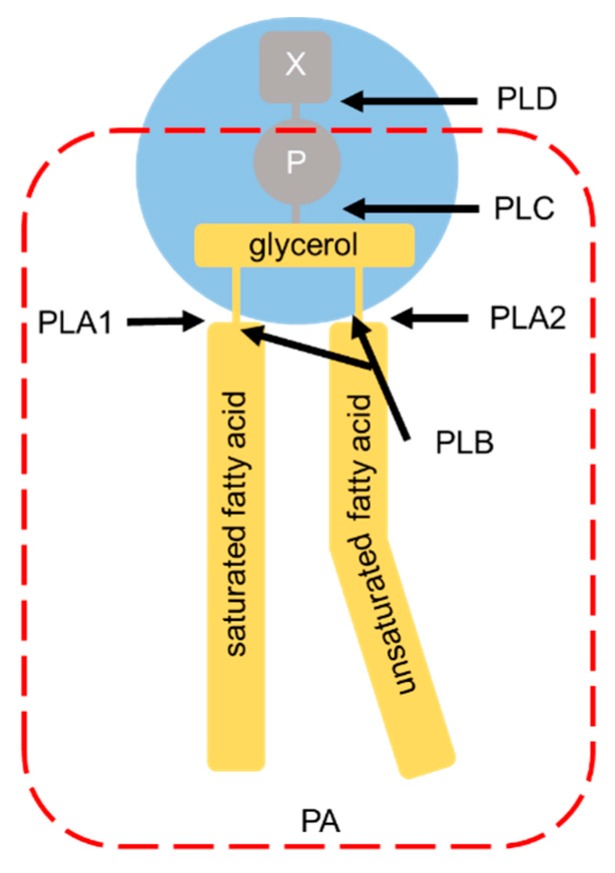
Phospholipase hydrolysis sites. Phospholipase classification is based on their specific sites of phospholipase-mediated hydrolysis. The head groups of the glycerophospholipids are bound to the glycerol moiety at the *sn*-3 position. X: choline (for phosphatidylcholine), ethanolamine (for phosphatidylethanolamine), serine (for phosphatidylserine), or myo-inositol (for phosphatidylinositol); P: phosphate group; PLD: phospholipase D; PLC: phospholipase C; PLA1: phospholipase A1; PLA2: phospholipase A2; PLB: phospholipase B. The arrows indicate the sites of hydrolysis by each phospholipase. The part surrounded by the red broken line corresponds to phosphatidic acid (PA).

**Table 1 ijms-21-03301-t001:** Binding affinity of the α-syn mutants to phospholipids compared with the wild-type α-syn.

Mutations	Affinity	Experiments	References
A30P A53T A30P	↓ → ↓	Detergent-resistant membranes purified from HeLa cells Detergent-resistant membranes purified from HeLa cells Detergent-resistant membranes purified from mouse brain	[51]
A30P A53T	↓ →	Liposomes: POPC/POPA (7:3 in molar ratio)	[54]
G51D	↓	Liposomes: DOPC/DOPE/DOPS (12:5:3 in molar ratio)	[86]
A30P A53T E46K	↓ → ↑	Liposomes: DOPC/DOPE/DOPS (2:5:3 in molar ratio)	[87]
A30P A53T E46K	↓ → ↑	Liposomes: POPC/POPS (1:1 in molar ratio)	[59]
E46K	↑	Liposomes: PC/PS/Cholesterol (7:7:6 in molar ratio)	[89]
A53T	→	Liposomes: POPC/POPS (1:1 in weight ratio)	[64]
A30P A53T	↓ →	Liposomes: POPC/POPS (1:1 in weight ratio)	[41]
H50Q	→	Liposomes: POPG and DOPC/DOPE/DOPS (12:5:3 in molar ratio)	[88]

Abbreviations: POPA, 1-palmitoyl 2-oleoyl-*sn*-glycero-3-phosphatidic acid; POPS, 1-palmitoyl-2-oleoyl-*sn*-glycero-3-phosphoserine; POPC, 1-palmitoyl-2-oleoyl-*sn*-glycero-3-phosphocholine; POPG, 1-hexadecanoyl-2-octadecenoyl-*sn*-glycero-3- phosphatidylglycerol; DOPS, 1,2-dioleoyl-*sn*-glycero-3-phosphoserine; DOPE, 1,2-dioleoyl-*sn*-glycero-3-phosphoethanolamine; DOPC, 1,2-dioleoyl-*sn*-glycero-3-phosphocholine. ↑, increased; ↓, decreased; →, unchanged.

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
