# Peer review of "Lipids: Key Players That Modulate α-Synuclein Toxicity and Neurodegeneration in Parkinson’s Disease"

_ijms, 2020, doi:10.3390/ijms21093301_

Round 1

Reviewer 1 Report

General Remarks

  1. The review is on the current and interesting subject.
  2. The article encompasses a wide range of information about the synuclein functions and mechanisms it is involved. It discusses the subject from different angles and multiple target points.
  3. The text is a bit chaotic and needs clear subjects segregation into particular paragraphs. The subjects seem to appear in random sequence. One paragraph should stem out of the other one, evolving in a story with gradually increasing number of details. First more general description of physiological functions of the protein (Introduction) following with in depth details and pathologic effects.
  4. The paragraph titles has little to do with their actual content. Please rearrange and/or rename.
  5. Pathological aspects should be clearly opposed to the physiological functions of synuclein and to the normal aging.
  6. Since the neurodegenerative diseases are in the title – the emphasis should be put also on them. The differential aspects of the particular neurodegenerative diseases pathomechanism involving synuclein should be indicated. What are differences and similarities between particular diseases mechanisms and synuclein role in them.
  7. Authors focus mostly on PD in the text. What about other synucleinopaties or mentioned in the Introduction MSA. If only PD is described the title should be changed.
  8. The “lipids” is the first word in the title while its synuclein that is the main issue described.
  9. Language is proficient.

Specific Remarks

  1. Abstract:
  2. Line 15 “Thus, the most effective target in terms of PD prevention is the  suppression  of  α-syn  conversion  from  the  functional  form  to  pathological  forms” It is not proven that targeting α-syn would be effective target for PD prevention. This statement is too strong and gives false impression. Please rephrase. For comparison in AD amyloid burden reduction was not effective disease treatment as for the prevention studies are ongoing but precaution must be taken.
  3. Line 17: “α-syn interacts  with  synaptic  vesicle  membranes  and  regulates neurotransmission.” This statement is also too strong giving impression that α-syn is the main regulator of neurotransmission. It was shown to be somehow involved in vesicles transport thus indirectly involved in DA release. Please rephrase.
  4. Abstract focuses on PD – it is not the only synucleinopathy with neurodegeneration. A change in the title could be advised if authors want to discuss mainly PD. Or the abstract should take into the account also other neurodegenerative synucleinopathies.

Introduction:

  1. Line 28: “Parkinson’s disease  (PD)  is  a  Lewy  body  (LB)  disease;  LB  diseases  are  characterized  by  the progressive accumulation of fibrillized α-synuclein (α-syn) in the affected regions” . I think synucleinopathy is more appropriate name.
  2. The introduction would gain clarity if started from explaining synucleinopathy, naming the diseases belonging to the group, giving general description of common denominator. Also taking under consideration the neurodegenerative part, as indicated in the title.
  3. Why among all synucleinopaties and neurodegenerative disorders PD and MSA should be mostly described in the Introduction? – please give a reason.
  4. The problem here discussed should be identified and the goal of the review clearly indicated in the Introduction.
  5. Line 49: “This review focuses on the literature linking the α-syn function (or misfunction) with phospholipids and membrane structures.” The title is “Lipids..” yet the goal states “phosphplipids”. Please adjust which is more adequate.

Paragraph 2. Properties of the α-syn protein

  1. Line 56: Please start from the tissue distribution of α-syn before describing in vitro. You mention it is in the brain - where else. Also in the neurons – is it present in the other cell types? To what extent. Some LB are also observed in glia in PD.
  2. Line 98: “zebra finch “
  3. The description of “Lipids” and their function, subtypes and variety should be earlier as this is the main subject of the review. Cholesterol, cardiolipin, monosialogangliosides, PUFAs etc. - What about them?
  4. Please describe the types of synuclein aggregation forms.
  5. Please gather together the mechanisms of synuclein cytotoxicity,
  6. The content of figures should be described in the text.
  7. The number of abbreviations should be minimised to ease text comprehension, for example RRP, AZ, RP. If it is not used at least 3-4 times please do not abbreviate. DLB is not explained. Abbreviation list should be sorted according to the alphabet.
  8. Line 165: “overexpression of A30P α-syn, which has been suggested to decrease its affinity to membranes [32], shows no defect in a synaptic vesicle exocytosis [57]. In contrast to A30P, A53T and E46K, which retain the  membrane-binding  affinity” synuclein mutations and posttranslational modifications should be introduced in the text earlier before discussing their effect. Furthermore experimental studies on genetically modified synuclein forms as well as KO, KI, or overexpression should be clearly separated in the text from the experiments with natural protein form.
  9. The new perspectives, gaps in the knowledge, potential treatments, new methods for studying the subject etc. should be also indicated at the end of the review.

Author Response

Response to Reviewer 1 Comments

We greatly appreciate the feedback from this reviewer on our manuscript (Manuscript ID: ijms-778143) entitled Lipids: Key players that modulate α-synuclein toxicity and neurodegeneration in Parkinson’s disease. We found the comments to be highly constructive for improving our manuscript. We have incorporated the reviewer’s suggestions and advice into the revised manuscript. Below are our point-by-point responses to the reviewers’ comments.

Specific Remarks

Abstract: 

Line 15 “Thus, the most effective target in terms of PD prevention is the suppression of α-syn conversion from the functional form to pathological forms” It is not proven that targeting α-syn would be effective target for PD prevention. This statement is too strong and gives false impression. Please rephrase. For comparison in AD amyloid burden reduction was not effective disease treatment as for the prevention studies are ongoing but precaution must be taken.

We appreciate this reviewer’s appropriate comment. We changed the sentence as follows:

“Thus, one of potential targets in terms of PD prevention is the suppression of α-syn conversion from the functional form to pathological forms.

Line 17: “α-syn interacts with synaptic vesicle membranes and regulates neurotransmission.” This statement is also too strong giving impression that α-syn is the main regulator of neurotransmission. It was shown to be somehow involved in vesicles transport thus indirectly involved in DA release. Please rephrase.

We appreciate this reviewer’s appropriate comment. We changed the sentence as follows:

Recent studies suggested that α-syn interacts with synaptic vesicle membranes and modulate the synaptic functions

Abstract focuses on PD – it is not the only synucleinopathy with neurodegeneration. A change in the title could be advised if authors want to discuss mainly PD. Or the abstract should take into the account also other neurodegenerative synucleinopathies.

In this review, we focused on the relationship between lipid and mainly PD. Thus, we would like to change the title as “Lipids: Key players that modulate α-synuclein toxicity and neurodegeneration in Parkinson’s disease”. Because we believe that other neurodegenerative synucleiopathies are also important to consider the pathogenesis by aggregated α-synuclein, we left the description of other synucleiopathies in the Introduction.

Introduction: 

Line 28: “Parkinson’s disease (PD) is a Lewy body (LB) disease; LB diseases are characterized by the progressive accumulation of fibrillized α-synuclein (α-syn) in the affected regions”. I think synucleinopathy is more appropriate name.

Thank you for your comment. We rephrased “LB disease” to “synucleinopathy”.

The introduction would gain clarity if started from explaining synucleinopathy, naming the diseases belonging to the group, giving general description of common denominator. Also taking under consideration the neurodegenerative part, as indicated in the title.

According to the suggestion, we modified the first paragraph of the Introduction as follows:

Several neurodegenerative diseases, including Parkinson’s disease (PD), PD dementia (PDD), dementia with Lewy bodies (DLB), and multiple system atrophy (MSA) are described as synucleinopathies, which are characterized by the progressive accumulation of fibrillized α-synuclein (α-syn) in the affected regions [1, 2]. Lewy bodies (LBs), which consist mainly of fibrillized α-syn, is the main pathological hallmark of PD, PDD, and DLB. PD is the most common synucleinopathy in the elderly.….

Why among all synucleinopaties and neurodegenerative disorders PD and MSA should be mostly described in the Introduction? – please give a reason.

LB pathology is a common feature in PD and other synucleinopaties such as PDD and DLB, whereas GCIs is the pathological hallmark of MSA and structurally distinct from LB. We believe that the underlying cause of aggregating process are different between LB and GCIs. We mentioned PD and MSA as typical diseases of LB and GCIs.  

The problem here discussed should be identified and the goal of the review clearly indicated in the Introduction.

We revised the introduction to clarify this point as follows:

Unfortunately, there is no definitive therapy for PD. Understanding the pathogenic mechanisms associated with α-syn aggregation is crucial to develop effective disease-modifying therapies [28]. This review focuses on the literature linking the α-syn function (or misfunction) mainly with phospholipids, which is a major constituent molecule of biomembrane, and membrane structures, and how these interactions participate in the physiological function and the pathogenesis in PD. Finally, we would like to discuss potential therapeutic targets for lipid modification in PD.

Line 49: “This review focuses on the literature linking the α-syn function (or misfunction) with phospholipids and membrane structures.” The title is “Lipids.” yet the goal states “phosphplipids”. Please adjust which is more adequate.

We changed “phosphplipids” to “lipids”.

Paragraph 2. Properties of the α-syn protein

Line 56: Please start from the tissue distribution of α-syn before describing in vitro. You mention it is in the brain - where else. Also in the neurons – is it present in the other cell types? To what extent. Some LB are also observed in glia in PD.

We added the information about the tissue distribution of α-syn as follows:

Although α-syn is ubiquitously distributed in red blood cells, lymphocytes, muscle, kidney, heart, and lungs [24-28], it is very abundant in the brain tissue, accounting for about 1% of the total protein in a rat brain [29].

Line 98: “zebra finch “

This study has done with small birds, finches.

The description of “Lipids” and their function, subtypes and variety should be earlier as this is the main subject of the review. Cholesterol, cardiolipin, monosialogangliosides, PUFAs etc. - What about them?

According to the reviewer’s suggestion, we added some sentences in the Introduction.

A variety of lipids exist in the brain, including fatty acids, triacylglycerol, phospholipid, sterol, and glycolipid. These lipid species are utilized for energy metabolism, protein modification, signal mediators, and biomembrane and organelle functions.…

…..This review focuses on the literature linking the α-syn function (or misfunction) mainly with phospholipids, which is a major constituent molecule of biomembrane, and membrane structures,….

Please describe the types of synuclein aggregation forms.

We assume this is a comment regarding the issue of α-syn stains. To our knowledge, there is no literature showing how lipids affect the determination of α-syn strains. 

Please gather together the mechanisms of synuclein cytotoxicity,

We added sentences as follows:

Many neurotoxic mechanisms mediated by α-syn in conjunction with lipids have been proposed. α-syn aggregation is triggered by the disruption of α-syn affinity to membrane [26], or induced by the pore-like morphology of α-syn on the membranes [118] while the membrane disruption results in increased intracellular reactive oxygen species and reduction in the mitochondrial activity [119]. Increasing membrane curvature and expansion by α-syn impairs membrane integrity [120, 121]. These models should be validated by further studies.

The content of figures should be described in the text. The number of abbreviations should be minimised to ease text comprehension, for example RRP, AZ, RP. If it is not used at least 3-4 times please do not abbreviate. DLB is not explained. Abbreviation list should be sorted according to the alphabet.

We revised those points, accordingly.

Line 165: “overexpression of A30P α-syn, which has been suggested to decrease its affinity to membranes [32], shows no defect in a synaptic vesicle exocytosis [57]. In contrast to A30P, A53T and E46K, which retain the membrane-binding affinity” synuclein mutations and posttranslational modifications should be introduced in the text earlier before discussing their effect. Furthermore, experimental studies on genetically modified synuclein forms as well as KO, KI, or overexpression should be clearly separated in the text from the experiments with natural protein form.

We added the terms “overexpression”, “native”, or “endogenous” in the experiments of each study.

According to the advice, we introduced the α-syn mutations earlier as follows:

There are six α-syn point mutations known to be associated with familial PD (A30P, E46K, H50Q, G51D, A53E, and A53T), which are in the N-terminal membrane-binding domain [11-15]. The overexpression of A30P….

The new perspectives, gaps in the knowledge, potential treatments, new methods for studying the subject etc. should be also indicated at the end of the review.

Perspectives and potential treatments are described in the Conclusions.

α-syn seeds for propagation are suggested to come from the intestinal nerve plexus, which should be subjected to lipid constituents in diet. Altered lipid metabolism changes the membrane property and could stimulate α-syn propagation through vesicular transport such as exsosome [153]. The determination of risk lipids in this context is a challenging problem in the future.…..

…. The concept of control of α-syn conformation by lipid regulation is a promising candidate for disease-modifying therapy in PD [28]. Indeed, an antimicrobial agent, squalamine [91], and a small molecule, NPT200-11 [155], which have advanced into a clinical trial (NCT02606682), successfully prevent α-syn aggregation by modulating α-syn-binding affinity to membranes. Squalamine from dogfish sharks was found to displace α-syn from the surface of lipid membrane by masking α-syn lipid-binding site. This molecule is reported to inhibit aggregation of α-syn [91]. The dietary manipulation of fatty acids over a long time period could change brain lipid components to prevent α-syn aggregation [26].

Reviewer 2 Report

The review by Mori et al., entitled “Lipids: Key players that modulate alpha-synuclein toxicity and neurodegeneration” is a thorough description of the field. It covers many aspects of how interactions of alpha-syn with lipids, including phospholipids, are necessary for physiological function and how an imbalance in that interaction can cause alpha-syn aggregation ultimately resulting in Parkinson’s pathogenesis.

The language is quite good in the manuscript although it is packed with information and might be a little dense for a non-expert. There are a few points that could improve the review.

  1. The first paragraph of the introduction mentions that alpha-syn is expressed in oligodendrocytes but the paper focuses on neurons. What is known about the role of alpha-syn in glia?
  2. Are there any in vitro studies that have been done with synaptosomes to mimic more closely the in vivo environment?
  3. Fig 3 does not have SNARE or some of the other synaptic vesicle proteins labeled which would help when reading the text.
  4. It is not quite clear how if overexpression of alpha-syn reduces vesicles in the pool by inhibiting the re-clustering of synaptic vesicles, how overexpression would rescue CSPalpha knockout mice by promoting SNARE complex assembly (page 5).   These concepts appear to be contradictory.
  5. Given the contradictory results of alpha-syn in vesicle docking, which result do the authors think is the more accurate one? Lines 183-184
  6. Why do the authors think there is a difference in the two studies with regard to mitochondria-associated endoplasmic reticulum membranes. A little more synthesis of published studies overall would benefit the reader. Lines 230-233
  7. It would be helpful to include some more information on the proposed mechanism by which the antimicrobial agent and small molecule are modulating alpha-syn affinity to membranes. This would relate the clinical trials back to the molecular pathways described in the rest of the review.
  8. Line 175 typo in chaperone
  9. Line 227 extra tab indent
  10. Line 327 typo in variety

Author Response

Response to Reviewer 2 Comments

We greatly appreciate the feedback from this reviewer on our manuscript (Manuscript ID: ijms-778143) entitled Lipids: Key players that modulate α-synuclein toxicity and neurodegeneration in Parkinson’s disease. We found the comments to be highly constructive for improving our manuscript. We have incorporated the reviewer’s suggestions and advice into the revised manuscript. Below are our point-by-point responses to the reviewers’ comments.

  1. The first paragraph of the introduction mentions that alpha-syn is expressed in oligodendrocytes but the paper focuses on neurons. What is known about the role of alpha-syn in glia?

The physiological roles of α-syn in glia is unknown. We added sentences as follows:

α-syn is expressed predominantly in neurons. However, α-syn forms aggregation in glial cells in these synucleinopathies, including PD [4, 5].….

….Although the physiological function of α-syn in glial cells is not fully understood,α-syn is suggested to activate the innate immune system, in which microglia and astrocytes are involved, and to regulate astrocytic fatty acid metabolism[9, 10].

  1. Are there any in vitro studies that have been done with synaptosomes to mimic more closely the in vivo environment?

Diao, et al eLife, 2013 (Native α-synuclein induces clustering of synaptic-vesicle mimics via binding to phospholipids and synaptobrevin-2/VAMP2) is one study. We cited it as ref .77.

  1. Fig 3 does not have SNARE or some of the other synaptic vesicle proteins labeled which would help when reading the text.

The well characterized SNARE proteins in terms of molecular association with α-syn are Synaxin-1A, SNAP25 and VAMP2. For simplicity, we depicted these three proteins although we appreciate this reviewer’s suggestion. SGT, CSPαand Hsc70 were also depicted in the figure.

  1. It is not quite clear how if overexpression of alpha-syn reduces vesicles in the pool by inhibiting the re-clustering of synaptic vesicles, how overexpression would rescue CSPalpha knockout mice by promoting SNARE complex assembly (page 5). These concepts appear to be contradictory.

The mechanism how the overexpression of α-syn rescues CSP-α knockout mice remains unknown. We removed the phrase “by promoting the SNARE complex assembly”.

  1. Given the contradictory results of alpha-syn in vesicle docking, which result do the authors think is the more accurate one? Lines 183-184

The in vitro study by Dr. Shin’s group (ref. 81) used oligomers generated by dopamine while Dr. Sudhof’s group (refs. 76, 77) employed the native forms of α-syn. Thus, the former could reflect the pathological roles of α-syn and the latter could reflect physiological roles of α-syn.

  1. Why do the authors think there is a difference in the two studies with regard to mitochondria-associated endoplasmic reticulum membranes. A little more synthesis of published studies overall would benefit the reader. Lines 230-233

We added mechanistical explanations as follows:

A cultured cell study showed that a wild-type α-syn overexpression increased the number of MAMs, leading to an increased mitochondrial Ca2+ uptake from the ER to sustain mitochondria function [99], and another study showed that A30P and A53T α-syn reduced its association with MAMs, promoting a mitochondrial fragmentation [100]. Mechanistically, A30P showed the lower localization in MAM because of its less affinity to the membrane, while A53T showed a reduction in both membrane binding and the total protein expression compared with wild-type [100].

  1. It would be helpful to include some more information on the proposed mechanism by which the antimicrobial agent and small molecule are modulating alpha-syn affinity to membranes. This would relate the clinical trials back to the molecular pathways described in the rest of the review.

We added a sentence describing to molecular mechanism of squalamine.

The concept of control of α-syn conformation by lipid regulation is a promising candidate for disease-modifying therapy in PD [28]. Indeed, an antimicrobial agent, squalamine [91], and a small molecule, NPT200-11 [155], which have advanced into a clinical trial (NCT02606682), successfully prevent α-syn aggregation by modulating α-syn-binding affinity to membranes. Squalamine from dogfish sharks was found to displace α-syn from the surface of lipid membrane by masking α-syn lipid-binding sites. This molecule is reported to inhibit aggregation of α-syn [91].

  1. Line 175 typo in chaperone
  2. Line 227 extra tab indent
  3. Line 327 typo in variety

We corrected them.

Reviewer 3 Report

In this review the authors would correlate the role of phospholipids and lipid metabolism in α-synuclein dynamics and aggregation properties. The topic is important, but several points are not analyzed in detail preventing the publication of the article in the present form. In general, this article does not seem so exhaustive on the role of lipids in α-synuclein toxicity and on development of neurodegeneration. In my opinion the authors should delve into the subject a little more.

In more detail:

The titles of the paragraphs are not well chosen and the word α-synuclein is written as such and as α-syn. Anyway, never use the abbreviations in the titles.

In the introduction, the authors should describe a little bit more the lipids and their role.

Then, they speak about acetylation of α-synuclein, but this PTM is not contextualized.

They introduce a hypothetical relation between mutations of proteins implicated in the lipid metabolism and PD, but the concept is not clear. Which proteins and which mutations....

Along the text, α-synuclein N-T (1-95) and NAC (61-95) are described as overlapping, but Figure 1 does not illustrate this point. Anyway, please define correctly the membrane domain in the protein.

α-Synuclein secondary structure upon interaction with membrane is touched on, but the native structure of α-synuclein is not considered and in any case the question of α-synuclein structure.

The issue about the correlation between α-synuclein membrane interaction and aggregation is not clear, and it is the central theme of the review (Paragraph 6).

And it is not clear how a lipid impairment leads to PD.

The style and the quality of figures must be improved.

Author Response

Response to Reviewer 3 Comments

We greatly appreciate the feedback from this reviewer on our manuscript (Manuscript ID: ijms-778143) entitled Lipids: Key players that modulate α-synuclein toxicity and neurodegeneration in Parkinson’s disease. We found the comments to be highly constructive for improving our manuscript. We have incorporated the reviewer’s suggestions and advice into the revised manuscript. Below are our point-by-point responses to the reviewers’ comments.

The titles of the paragraphs are not well chosen and the word α-synuclein is written as such and as α-syn. Anyway, never use the abbreviations in the titles.

Thank you for the suggestion. We corrected the titles.

In the introduction, the authors should describe a little bit more the lipids and their role.

According to the reviewer’s suggestion, we added some sentences in the Introduction.

A variety of lipids exist in the brain, including fatty acids, triacylglycerol, phospholipid, sterol, and glycolipid. These lipid species are utilized for energy metabolism, protein modification, signal mediators, and biomembrane and organelle functions….

…..This review focuses on the literature linking the α-syn function (or misfunction) mainly with phospholipids, which is a major constituent molecule of biomembrane, and membrane structures,….

Then, they speak about acetylation of α-synuclein, but this PTM is not contextualized.

We added a sentence as follows:

“Moreover, it has been shown that post-translational modifications of α-syn are another factor to modulate its lipid binding.”

They introduce a hypothetical relation between mutations of proteins implicated in the lipid metabolism and PD, but the concept is not clear. Which proteins and which mutations....

To clear the concept, we added sentences as follows:

The findings of PLA2G6, VPS13C, LIMP2, GBA1, and GALC as genes responsible for or genetic risks of PD with prominent LB deposition strengthen the concept that lipids are indeed involved in the aggregation and propagation of α-syn [21-25]. These genes are suggested to regulate metabolism, transport and degradation of lipids [22, 23, 25-27].

Along the text, α-synuclein N-T (1-95) and NAC (61-95) are described as overlapping, but Figure 1 does not illustrate this point. Anyway, please define correctly the membrane domain in the protein.

We corrected the-N-terminal repeat region as residues 1-60.

α-Synuclein secondary structure upon interaction with membrane is touched on, but the native structure of α-synuclein is not considered and in any case the question of α-synuclein structure.

We added images to explain the conformational change of α-syn in the presence or absence of biomembrane in Fig 2.

The issue about the correlation between α-synuclein membrane interaction and aggregation is not clear, and it is the central theme of the review (Paragraph 6).

We added sentences to explain the proposed mechanisms as follows:

“However, many neurotoxic mechanisms mediated by α-syn in conjunction with lipids have been proposed. α-syn aggregation is triggered….”

And it is not clear how a lipid impairment leads to PD.

It is not fully understood how lipid alteration leads to PD and this is the very reason we focus on this issue. We discuss this issue, introducing literature reporting the relationship between α-syn and lipids and neurodegeneration caused by its disruption.

The style and the quality of figures must be improved.

Although this comment is not specific, we improved Fig. 2 to explain the conformational change of α-syn in the presence or absence of biomembrane. We also changed the color of α-syn in Fig. 1 to match the molecular model in Fig. 2.

Round 2

Reviewer 1 Report

The text was improved.

Further corrections required:

Throughout the whole text – please focus on insertion of the literature citations. For example after the strong statements like this:

Line 31: “Lewy bodies (LBs), which consist mainly of fibrillized α-syn,  is  the  main  pathological  hallmark  of  PD,  PDD,  and  DLB.”

Line 32: “PD  is  the  most  common synucleinopathy in the elderly.”

Line 78 “The  brain is one of the richest in lipid content among the body tissues.”

etc. Look through the whole text and fill the gaps.

Check the whole text for abbreviations – some of them are not explained – for example line 83: “soluble NSF-attachment”

Line 105: “α-syn is divided into three overlapping regions including (1) the-N-terminal repeat region (residues 1-60), (2) the highly hydrophobic central portion (residues 61-95), and (3) the acidic C-terminal region (residues 96-140).” 1-60, 61-96, 96-140 – how is it overlapping?

Rephrase the subtitles into the sentence: for example

“3. N-terminal amphipathic helices of for phospholipid binding” à ‘N-terminal amphipathic helices of α-synuclein bind with phospholipids’ or in analogy to the other titles ‘Phospholipid binding properties of α-synuclein N-terminal amphipathic helices’

Line 232: “neutral  phospholipid  PC “ what PC stands for? Explain the abbreviation.

Line 471: “retromer” - what is it

Subtitle “Other lipid metabolisms “ please rephrase – metabolism is a general term for multiple mechanisms. It should not be plural.

Line 506: “α-syn  seeds  for  propagation  are  suggested  to  come  from  the intestinal  nerve  plexus,  which  should  be  subjected  to  lipid  constituents  in  diet.” This sentence needs rephrasing to clarify.  Synuclein seeds? What should be subjected to what?

Also line 507: “Altered  lipid metabolism  changes  the  membrane  property” maybe better ‘Altered  lipid metabolism  affects/influences  the  membrane  property’

Authors write in conclusions “In  the  brain,  in  addition  to  the  anterior  cingulate  cortex  and amygdala, a variety of lipid species change in the visual cortex where the LB pathology is rarely observed, indicating that changes in lipids precede an LB deposition [154].”

Since Authors wanted to focus on PD

  • Are there any known changes in lipid composition or metabolism in the regions affected by neurodegeneration such as the human substantia nigra or striatum of patients, or related to normal aging? Please include such information in the text.
  • Does changes in lipid metabolism localized in different structures of the human brain correlate with localization of Lewy bodies? Please comment in the text.

Author Response

Point 1. Throughout the whole text – please focus on insertion of the literature citations. For example, after the strong statements like this:

Line 31: “Lewy bodies (LBs), which consist mainly of fibrillized α syn, is the main pathological hallmark of PD, PDD, and DLB.”

Line 32: “PD is the most common synucleinopathy in the elderly.”

Line 78 “The brain is one of the richest in lipid content among the body tissues.”

etc. Look through the whole text and fill the gaps.

- We cited appropriate references for the sentences you pointed out and other sentences.

Point 2. Check the whole text for abbreviations – some of them are not explained – for example line 83: “soluble NSF-attachment”

- We replaced “soluble NSF-attachment” with “the soluble N-ethylmaleimide-sensitive factor-attachment” and check the whole text for abbreviations.

Point 3. Line 105: “α-syn is divided into three overlapping regions including (1) the-N-terminal repeat region (residues 1-60), (2) the highly hydrophobic central portion (residues 61-95), and (3) the acidic C-terminal region (residues 96-140).” 1-60, 61-96, 96-140 – how is it overlapping?

- We removed the word “overlapping”.

Point 4. Rephrase the subtitles into the sentence: for example

“3. N-terminal amphipathic helices of for phospholipid binding” à ‘N-terminal amphipathic helices of α-synuclein bind with phospholipids’ or in analogy to the other titles ‘Phospholipid binding properties of α-synuclein N-terminal amphipathic helices’

- Thank you for the suggestion. We modified the subtitle as follows:

“3. N-terminal amphipathic helices of α-synuclein bind with phospholipids”

Point 5. Line 232: “neutral  phospholipid  PC “ what PC stands for? Explain the abbreviation.

- We corrected it as follows:

“neutral phospholipid phosphatidylcholine (PC)”

Point 6. Line 471: “retromer” - what is it

- We added a phrase to explain the retromer as follows:

the retromer (a protein complex involved in the retrograde transport of membrane proteins in the endosomal-Golgi recycling pathway)

Point 7. Subtitle “Other lipid metabolisms “ please rephrase – metabolism is a general term for multiple mechanisms. It should not be plural.

- We rephrased it as “Other lipid-related genes”.

Point 8. Line 506: “α-syn  seeds  for  propagation  are  suggested  to  come  from  the intestinal  nerve  plexus,  which  should  be  subjected  to  lipid  constituents  in  diet.” This sentence needs rephrasing to clarify.  Synuclein seeds? What should be subjected to what?

- We modified the sentences as follows:

α-syn seeds for propagation are suggested to come from the intestinal nerve plexus. The transition of α-syn to seeding should be subjected to lipid constituents in diet.

Point 9. Also line 507: “Altered  lipid metabolism  changes  the  membrane  property” maybe better ‘Altered  lipid metabolism  affects/influences  the  membrane  property’

- Thank you for the suggestion. We replaced “changes” with “affects”.

Point 10. Authors write in conclusions “In the brain, in addition to the anterior cingulate cortex and amygdala, a variety of lipid species change in the visual cortex where the LB pathology is rarely observed, indicating that changes in lipids precede an LB deposition [154].”

Since Authors wanted to focus on PD

  • Are there any known changes in lipid composition or metabolism in the regions affected by neurodegeneration such as the human substantia nigra or striatum of patients, or related to normal aging? Please include such information in the text.
  • Does changes in lipid metabolism localized in different structures of the human brain correlate with localization of Lewy bodies? Please comment in the text.

- According to your advice, we added sentences as follows:

In the substantia nigra of PD patients, male-specific reduction in PC, PE and PI species and elevated activities of phospholipid biosynthetic enzymes were reported [155, 156] whereas reduction in GCase activity was observed [145]. These observations might be associated with dopaminergic neurodegeneration and LB pathology.